# The discovery of an *in situ* Neanderthal remain in the Bawa Yawan Rockshelter, West-Central Zagros Mountains, Kermanshah

Saman Heydari-Guran[1,2,3]*, Stefano Benazzi[4,5], Sahra Talamo[5,6], Elham Ghasidian[1,2], Nemat Hariri[2,7], Gregorio Oxilia[4], Samran Asiabani[3,8], Faramarz Azizi[3], Rahmat Naderi[3], Reza Safaierad[9], Jean-Jacques Hublin[5,10], Robert A. Foley[11], Marta M. Lahr[11]

1 Stiftung Neanderthal Museum, Mettmann, Germany, 2 Department of Prehistoric Archaeology University of Cologne, Cologne, Germany, 3 DiyarMehr Centre for Palaeolithic Research, Kermanshah, Iran, 4 Department of Cultural Heritage, University of Bologna, Bologna, Italy, 5 Department of Human Evolution, Max Planck Institute for Evolutionary Anthropology, Leipzig, Germany, 6 Department of Chemistry G. Ciamician, Alma Mater Studiorum, University of Bologna, Bologna, Italy, 7 Department of Archaeology, University of Mohaghegh Ardabili University, Ardabil, Iran, 8 Department of Architecture, Faculty of Art and Architecture, Bu-Ali Sina University, Hamedan, Iran, 9 Department of Physical Geography, University of Tehran, Tehran, Iran, 10 Collège de France, 11 Place Marcelin Berthelot, Paris, France, 11 Leverhulme Centre for Human Evolutionary Studies, Department of Archaeology, University of Cambridge, Cambridge, United Kingdom

* heydari-guran@neanderthal.de

**Data Availability Statement:** All relevant data are within the paper.

## Abstract

Neanderthal extinction has been a matter of debate for many years. New discoveries, better chronologies and genomic evidence have done much to clarify some of the issues. This evidence suggests that Neanderthals became extinct around 40,000–37,000 years before present (BP), after a period of coexistence with *Homo sapiens* of several millennia, involving biological and cultural interactions between the two groups. However, the bulk of this evidence relates to Western Eurasia, and recent work in Central Asia and Siberia has shown that there is considerable local variation. Southwestern Asia, despite having a number of significant Neanderthal remains, has not played a major part in the debate over extinction. Here we report a Neanderthal deciduous canine from the site of Bawa Yawan in the West-Central Zagros Mountains of Iran. The tooth is associated with Zagros Mousterian lithics, and its context is preliminary dated to between ~43,600 and ~41,500 years ago.

## Introduction

Neanderthals were a very successful hominin lineage that existed for several hundred thousand years, and their extinction remains one of the most persistent questions in palaeoanthropology. With such a vast geographical range, which at times extended from westernmost Europe and across a very large area of Asia [1, 2] identifying the last surviving Neanderthal populations is critical for interpreting the mechanisms behind their demographic decline, the conditions that enhanced or reduced their resilience, and the geography and timing of their last interactions with modern humans. Unsurprisingly, one of the most controversial aspects of

**Funding:** Funding was provided from Deutsche Forschungsgemeinschaft (Grant 423897519) to SHG.

**Competing interests:** The authors have declared that no competing interests exist.

this debate is the chronology of their disappearance. Aspects of that chronology are relatively well established, in Europe, where Neanderthals appear to have a continuous occupation since the Middle Pleistocene, their last occurrence has been timed to ~ 40,000–37,000 cal. BP [1–4]. A number of sites testify to the presence of *Homo sapiens* in Europe from at least 46,000 BP—Bacho Kiro [1] and Grotta del Cavallo [5, 6]. Therefore, the chronology of Neanderthal extinction across most, if not all of Europe, implies a minimum of 4,000 years of contemporaneity with *Homo sapiens* groups [1, 2], during which both biological [7] and cultural [8] interactions took place. At the other extreme of their geographical range, the Neanderthal occupation of parts of Siberia reveals a very different pattern. Palaeoanthropological and ancient genomic evidence from Denisova, Chagyrskaya and Okladnikov caves suggests repeated west-to-east dispersals between 150,000 and 120,000 years [9–11] which gave rise to at least two temporary Neanderthal populations that became demographically vulnerable, characterised by episodes of interbreeding with local Denisovans and small population sizes [11–13].

However, there is one region that presents a particular challenge because of the sheer complexity of both its prehistoric record and its geographic setting—southwest Asia. Bordered by the Mediterranean Sea in the west and mountain ranges in the north and east, the core of the region consists of deserts and semi-deserts that, even during humid climate intervals, would have constrained resources along waterways [14]. Thus, the prehistoric record of southwest Asia is concentrated along an arc extending from the Levant in the west to the Caucasus in the north and the Zagros Mountains to the east. These areas—Levant, Caucasus and Zagros—in turn, offered hominins different challenges and opportunities, and this is reflected in their different records.

Biogeographically, the Levant oscillated between African and Eurasian biomes, acting as a temporary corridor for dispersing African populations [15]. It is now known that this corridor was used by *Homo sapiens* at least twice—between 177,000–194,000 years, as evidenced at the site of Misliya [16] and between ~120,000 and 90,000 years, as shown at the sites of Skhul and Qafzeh [17], before the area was permanently occupied by *Homo sapiens* around 55,000 years [18]. These intermittent early modern human occupations of the Levant were interspersed by the use of the area by Neanderthals [19], whose local extinction is represented by the last Middle Palaeolithic (MP) levels at Kebara at 48,000–49,000 BP [20]. However, the presence of *Homo sapiens* at Manot Cave ~55,000 years [18] suggests a relatively long period of overlap between the two groups in the area.

In contrast, the Caucasus, a 1200 km mountain range between the Black and Caspian Seas, was an important barrier to animal and hominin dispersals north, as shown by the genetic differentiation of several taxa into southern and northern Caucasus populations, and also reflected in differences in lithic technology [21,22]. Southern Caucasus MP assemblages appear to have been part of a Levantine network [21] that extended to the Zagros during the terminal MP phases [23]. However, in contrast to the Levant, Neanderthals occupied the Caucasus for nearly 10,000 years longer, with an age for the latest Neanderthal and MP industries similar to that of European Neanderthals, at ~ 40,000 cal. BP [22]. Yet, differently from both the Levantine and European records, *Homo sapiens* occupation of the Caucasus does not appear to occur prior to the local extinction of Neanderthals, thus suggesting no local contemporaneity of the two groups [22]. Ultimately, another picture of Neanderthal-*Homo sapiens* landscape interactions emerged from the Zagros Mountains which stretch over 1600 km in a northwest-southeast direction and reaching 4400 m above sea level (asl) in Iran. Moreover, due to the steep and rugged topographic conditions, the Zagros embrace diverse ecological ecotones. They represent a formidable geographical barrier, but also a diverse set of habitats and eco-zones [24] that were exploited by hominins since the Lower Palaeolithic [25]. MP sites are relatively numerous, although only three have yielded Neanderthal fossils. The best-known of

these is Shanidar, where the remains of ten Neanderthals were discovered [26–28]. The Shanidar Neanderthals have played a major role in the discussions about the complexity of Neanderthal behaviour, both for the presence of older-aged individuals in the fossil assemblage and the controversial claims of a 'flower burial'. Yet, despite their importance, their timing and contemporaneity have not been finely resolved; a broad age between 70,000 and 45,000 years has usually been reported, although recent excavations have suggested an age range of 55,000–45,000 years [28]. Approximately 350 km to the southeast, among a number of MP and UP sites, Wezmeh and Bisetun Caves in the Kermanshah region (Fig 1) have also produced Neanderthal remains. Wezmeh Cave has no artefactual evidence for human occupation in the Late Pleistocene, and the presence of a human premolar tooth (Wezmeh 1), recently shown to have Neanderthal affinities [29], was interpreted as the result of carnivore hunting/scavenging activities. The age estimation for this premolar is very poor, bracketed between 70,000 and 11,000 years based on the uranium-series analyses of the fauna by alpha spectrometry [29].

Nevertheless, the transitional period from Yafteh, Ghār-e Boof and Shanidar cave is bracketed between 45,100 and 40,350 cal. BP, with a 68.2% probability [30, 31]. Together with the approximate youngest age of the Neanderthal remains from Shanidar at 45,000 BP, this would suggest a pattern similar to the southern Caucasus, where modern humans succeed, rather than overlap in time with the local Neanderthals. Moreover, a probable, but undated, Neanderthal right proximal radius diaphysis was found in Bisetun Cave in association with typical Zagros Mousterian lithic artifacts [31, 32].

Despite these few Neanderthal fossil hominin remains, there is a widely held assumption that the MP assemblages in the Zagros Mountains were produced by Neanderthals, and that the local Middle to Upper Palaeolithic transition represents the arrival in the area of *Homo sapiens* [33]. Here we report the discovery of a new Neanderthal remain at Bawa Yawan rockshelter (BY1) in the Central Zagros Mountains confirming that Neanderthals manufactured Middle Palaeolithic artifacts. The new chronology associate with the site in question is showing a surprising continuity of Neanderthals presence in the area long after the first Upper Palaeolithic *Homo sapiens*, however, more radiocarbon dates are on the way to strengthen our conclusions.

## Materials and methods

All necessary permits for this research, including excavations and materials study to Saman Heydari-Guran, were issued by the Iranian Center for Archaeological Research (Permit number 963141/34/3952). The archaeological materials presented in this research are stored and available for the study with authorization from the Iranian Cultural Heritage and Tourism Organization (ICHTO) in Kermanshah Province.

The results for this study are based on the three excavation seasons between 2016–2018 in the archaeological site of Bawa Yawan (34˚ 38' 23.70"N, 46˚ 55' 48.36"E, 1300 m asl), locates in Kermanshah Region, Iran.

### µCT images of BY1

"High-resolution µCT images of the lower deciduous canine BY1 were obtained with the following scan parameters: 125 kilovoltage (kVp), 120 µA, and 0.015 mm voxel size. The µCT volume was segmented using Avizo 9.2 software (Thermo Fisher Scientific, Waltham, Massachusetts, US), and the 3D models of the dental tissues (i.e., enamel, dentine and the preserved portion of the pulp chamber) were refined in Geomagic Design X (3D Systems Software, Rock Hill, South Carolina, US) to optimize the triangles and create fully closed surfaces.

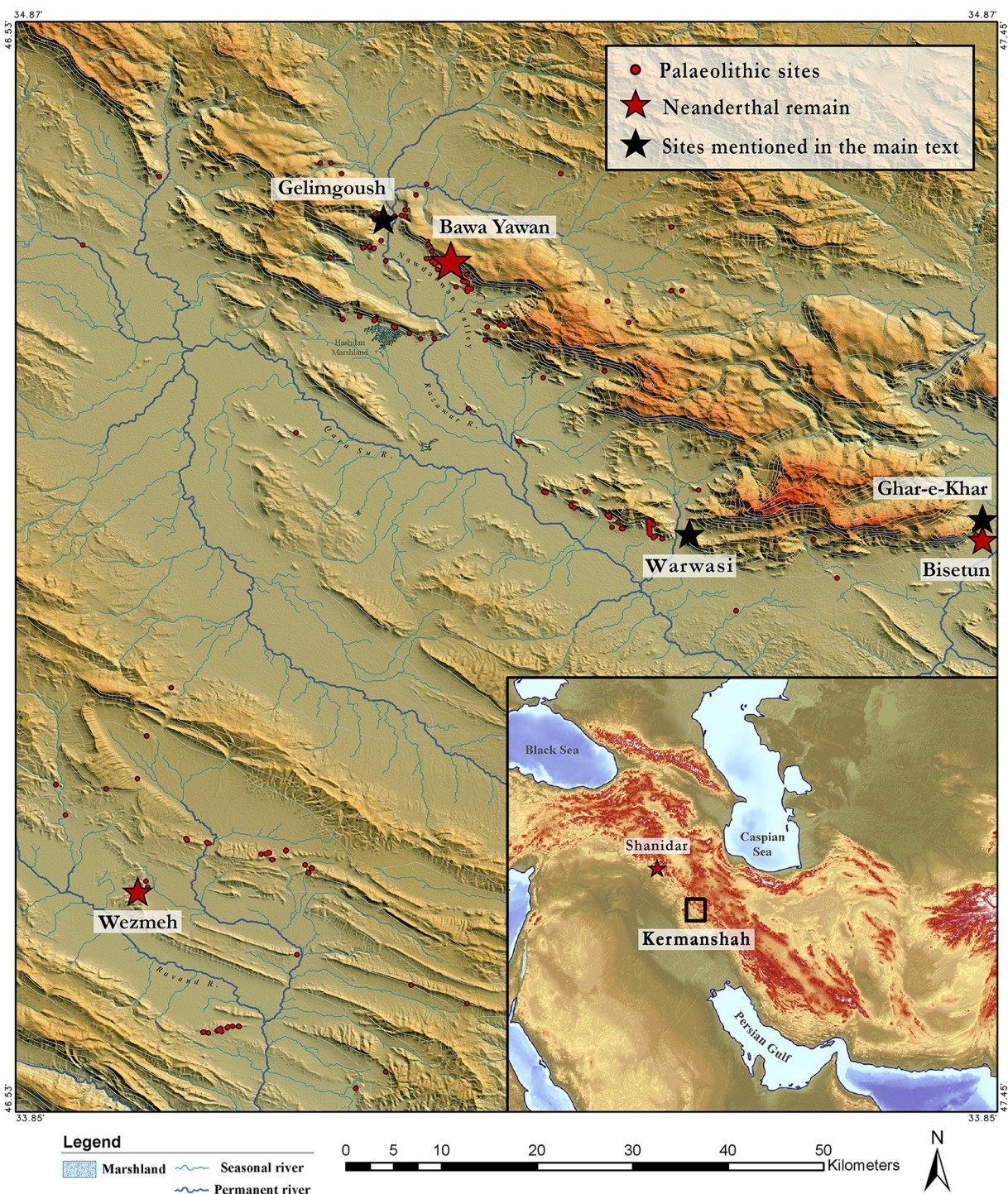

**Fig 1. The inset map shows the location of the Kermanshah region in the Zagros Mountain.** The main map shows the topography and hydrography of the Kermanshah region, the Bawa Yawan rockshelter in the Nawdarwan valley and the important sites mentioned in the text.

The morphological description of BY1 was performed according to standards outlined by the (ASUDAS [34]) Although ASUDAS has been devised for modern human permanent dentition, we applied protocol to fossil deciduous tooth as already done in other publications [e.g., 35–37] because it permits a more precise and accurate comparison at each degree of

development. The wear stage of the occlusal surface was assessed based on [38], while the age of root resorption was estimated according to Moorrees et al [39, 40]. The crown diameters of BY1 were virtually measured in Geomagic Design X, orienting the tooth with the best-fit plane computed at the cervical line parallel to the xy-plane of the Cartesian coordinate system, and with the lingual aspect parallel to the x-axis. The size of the bounding box enclosing the crown was used to collect mesiodistal (MD) and buccolingual (BL) diameters.

Comparative data for the BL diameter was collected from the literature [40–42]. The computation of the 3D average enamel thickness (AET) and relative enamel thickness (RET) index follow [41]. A set of Neanderthals (NEA; n = 6), early *H. sapiens* (EHS; n = 2), and recent *H. sapiens* (RHS; $n$ = 3) lower deciduous canines at different wear stages was acquired using a Skyscan scanner with voxel size ranging from 13 μm to 74 μm) at the Department of Human Evolution of the Max Planck Institute for Evolutionary Anthropology. As for BY1, dental tissues were segmented in Avizo 9.2 software, and the digital models were refined in Geomagic Design X. The 3D AET index (in mm) is the enamel volume divided by the underlying EDJ surface, while the 3D RET index (scale-free) is the 3D AET divided by the cube root of the crown dentine and pulp volumes.

## Radiocarbon dating

Thirteen charcoal samples collected from GH 5, 4, 3, and 2 were submitted for $^{14}$C dating to the Klaus-Tsichira-AMS facility of the Curt-Engelhorn Centre in Mannheim, Germany. Prior to dating, the samples were pretreated with the ABOX method and dated [43] in the radiocarbon lab. For samples that yielded finite conventional ages, calendar-year ages (and the corresponding 68.2% and 95.4% confidence intervals) were estimated using the IntCal20 calibration dataset [44].

## Results

### Nawdarwan Valley, Bawa Yawan rockshelter and stratigraphy

Located in the Nawdarwan Valley (34˚ 38' 23.70"N, 46˚ 55' 48.36"E, 1300 m asl), the Bawa Yawan (BY) rockshelter was discovered in 2009–10, during surveys for Palaeolithic sites in the Kermanshah region of the Central Zagros [33]. The Nawdarwan Valley forms a natural 17 km long corridor around the Razawar River that connects the two fertile plains of Kermanshah and Kamyaran in a N-S direction (Fig 1). The valley is rich in MP and UP rockshelters, with more than 50 sites identified during the surveys [33]. One of these, the Bawa Yawan rockshelter, is set in 50 m high limestone cliffs, visible from a far distance, that stand at the edge of a flat plain to the southwest (Fig 2A and 2B). A karstic spring, forming a natural pond, is located 50 m from the site. The archaeological deposits are accumulated on the southern slope of the cliff, occurring on a flat occupational area of ca. 300 m$^2$, which is 10 m higher than the plain surface today. Several large boulders lie at the eastern corner of the rockshelter and on the talus slope, protecting the archaeological deposits from erosion (Fig 3A). Excavations of the Bawa Yawan rockshelter took place during three field seasons between 2016–2018. These focused on two areas, 10 m apart, both located close to the rockshelter wall, named the Western (14 m$^2$) and Eastern (6 m$^2$) trenches (Fig 3A).

The Bawa Yawan deposits are divided into 5 geological horizons (GH 1–5) based on colour and sediment texture (Fig 3C). These have a gradient of about 10˚ from the back wall towards the outside, but are relatively level in east and west directions (Fig 3A and 3B). The deposits are mostly composed of clastic materials, and fine sediments washed into the site through joints and bedding planes from the back-wall cliff. In some squares (H31, I31) the excavation reached a maximum depth of 4.5 m from the surface without reaching bedrock (Fig 3B and 3C). Some

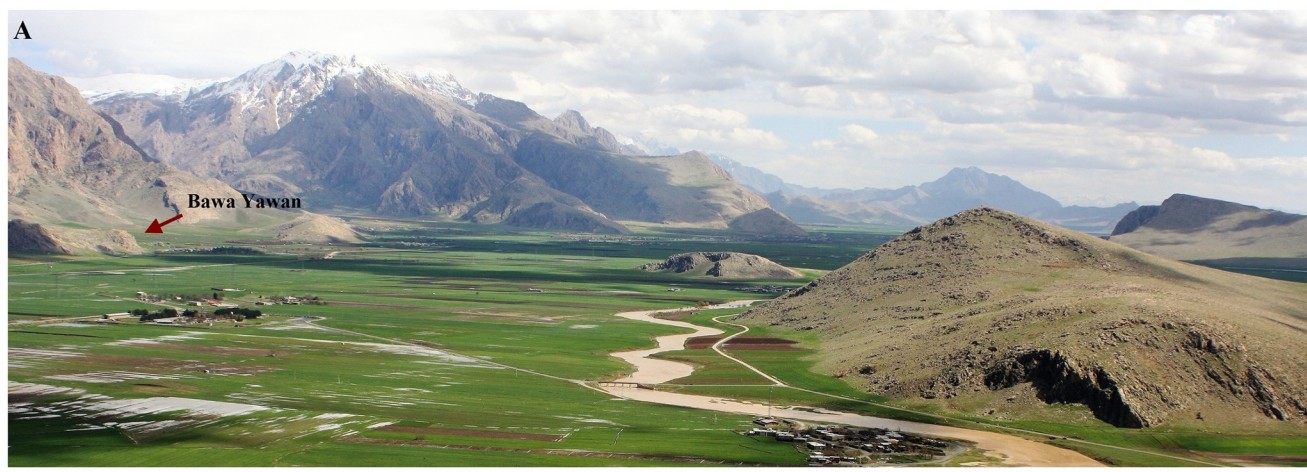

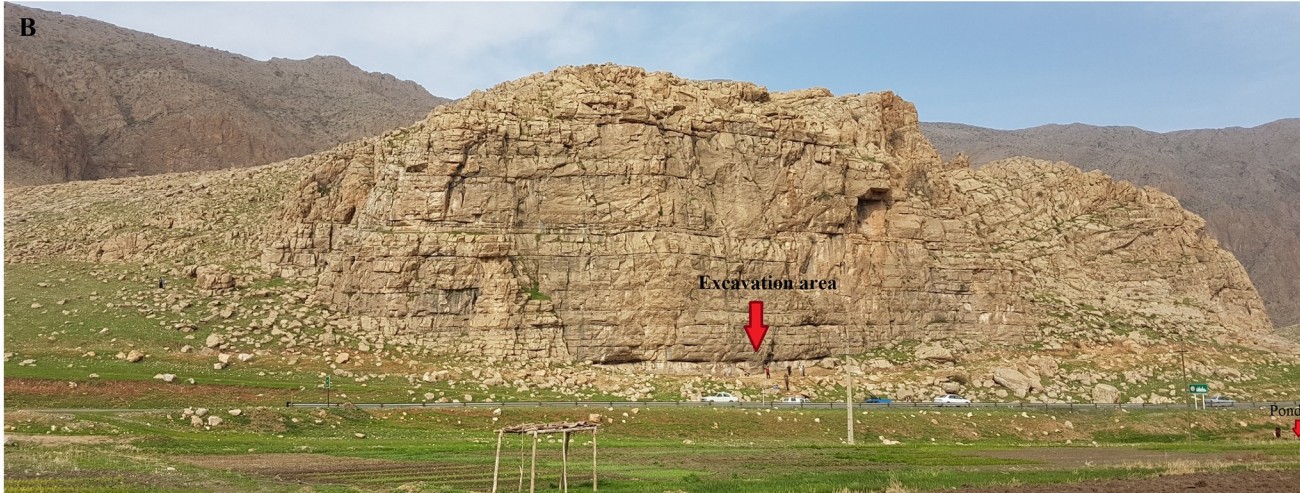

**Fig 2.** A. Picture shows the Nawdarwan Valley and red arrow indicates the position of the Bawa Yawan rockshelter. B. Picture shows the Bawa Yawan cliff and excavation area.

horizons, particularly GH2 and GH5, are characterised by a significant number of irregular stones collapsed from the cliff, varying in size from large boulders to small pebbles. GH3, with an average thickness of 80 cm, is different from the other geological horizons in being harder and having a higher carbonate content, protecting the underlying deposits from erosion.

- **GH5**: a dark reddish-brown (dry 5YR3/4 and moist 5YR2/4 very dark reddish-brown) and only exposed in five squares at Western trench and with 150 cm includes the lowermost geological layer and the deepest part of the excavation (Fig 3B and 3C). The sediments' colour becomes darker containing a lot of iron oxide particles, specifically in the lower part of the section. The deposits range from sandy/silty granules up to angular limestone of different sizes from small pebbles to boulders coming from the back wall of the shelter. The contact between GH4 and GH5 is gradual. The sediments become gradually reddish-dark brown downwards, specifically in the lowermost part of the section. It is the richest layer throughout the stratigraphy and is associated with numerous bone fragments which are partly burnt, very small charcoal fragments, and MP lithic artifacts of the Zagros Mousterian tradition. This GH yielded a good amount of fauna including *Bovidae*, *Bovidae antilopinae* and

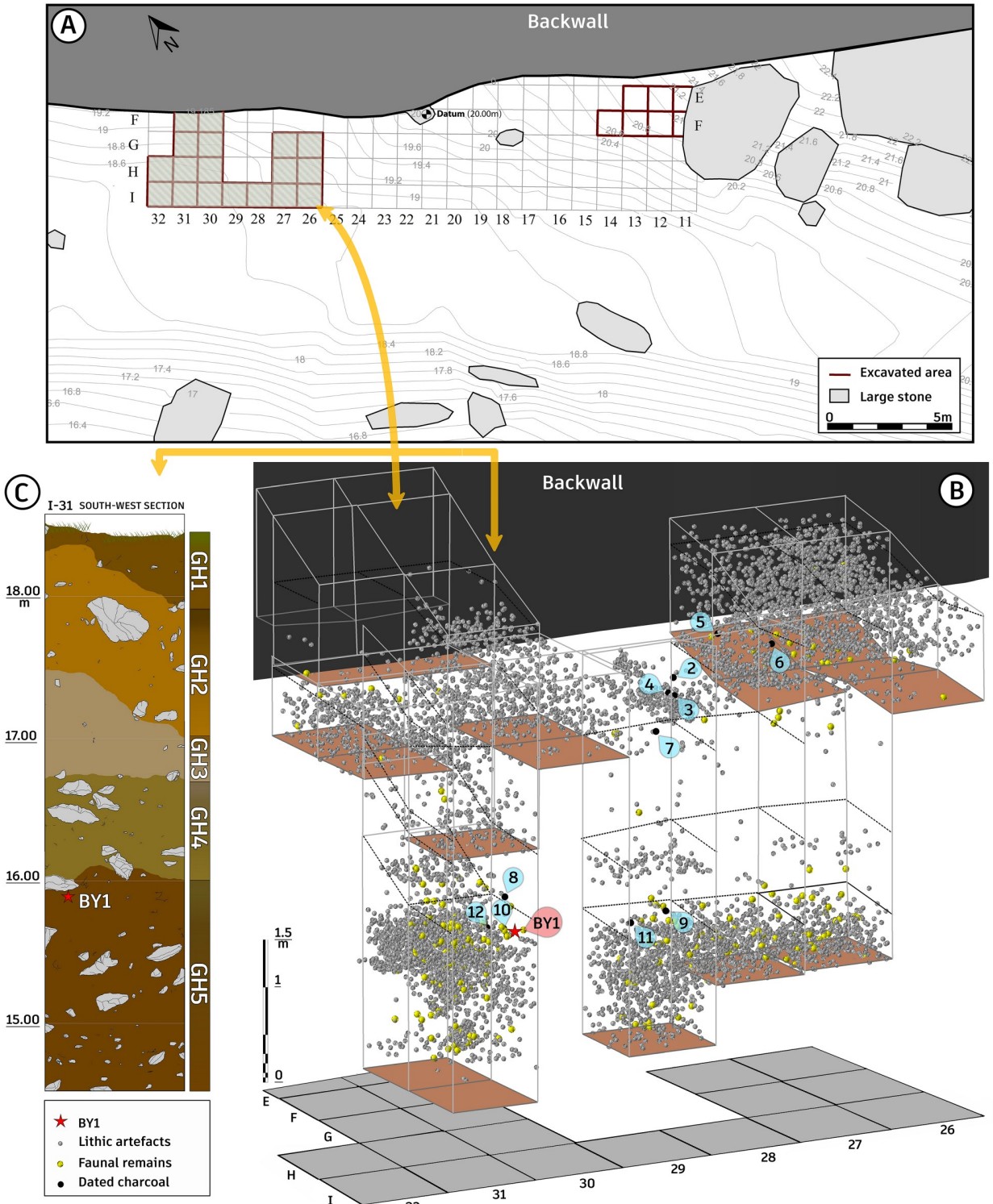

**Fig 3. Bawa Yawan rockshelter.** A. Topographic map and plan of the rockshelter showing the excavation area. B. 3D of the Western trench including position of BY1, dated charcoals, fauna and lithic artifacts. C. Profile shows the south-western wall of sq. I31, geological horizons and position of BY1.

*Equidae*. So far, bedrock has not been reached during the three excavation seasons and at a depth of 4.70 m (Fig 3).

- **GH4**: This GH with about 65 cm thickness and a light reddish-brown colour (dry 5YR4/6 reddish-brown and moist is 5YR7/3 dull orange) stands in sharp contrast with the underlying horizon (GH5). The amount of angular cobbles and boulders is more than the upper GHs. This GH faces a dramatic decrease in cultural materials, indicating a sudden abandonment of the site. The few lithic artifacts recovered in this layer belong to MP period (Fig 3).

- **GH3**: Around 95 cm thick, the GH3, changes relatively gradually from GH4 (dry 5YR7/3 dull reddish-brown and moist 5YR ¾ dark reddish-brown). It contains light colouartr compacted highly calcareous deposits with concentration of eboulis and angular pebbles. The number of lithic artifacts drops dramatically in this GH. Most of them are typical MP artifacts, some of which indicate Levallois technology. Very few UP artifacts have been also observed but only at the uppermost of GH3. The faunal remains are scattered and very fragmented as well (Fig 3B and 3C).

- **GH2**: This layer was observed in both Western and Eastern trenches. Up to 80–95 cm thick, GH2 is in general composed of reddish-brown deposits (dry 2.5YR5/4, moist 2.5YR2.5/3-dark reddish-brown) mixture of sand-silt-clay grain sediments associated with cobbles and medium-sized pebbles. This layer yielded archaeological materials from three different periods of MP, UP and Epipaleolithic (Fig 3B and 3C). GH2 yielded relatively abundant fauna, but highly fragmented including *Bovidae*, *caprinae*, and small birds.

- **GH1**: The uppermost geological layer is dull reddish-brown (dry 5Y6/4 dull orange, moist 2.5YR4/4, dullreddish-brown) and is around 30 cm thick, containing the recent soil associated with a few mixed modern objects and it is separated by an erosional unconformity to the underlying layer (GH2).

**BY1 Neanderthal tooth.**   At a depth of around 2.50 m from the surface in the upper part of GH5, where the density of MP cultural materials gradually reduced, an *in situ* hominin tooth (hereafter called BY1) was recovered in association with fauna and Zagros Mousterian lithic artifacts (Fig 3B and 3C). The tooth is an exfoliated lower left deciduous canine (Ldc1) consisting of a relatively well-preserved crown and about one-fourth of the root (Fig 4). Neither caries nor enamel hypoplasia is visible. The enamel shows several longitudinal fractures from the cervix to the incisal surface that affect the underlying dentine. The incisal surface is worn obliquely, mesially to distally, exposing a large area of dentine up to wear stage 4 [38]. The buccal surface exhibits a mesiodistal convexity with its maximum at the mesial aspect (Arizona State University Dental Anthropology System (ASUDAS) grade 4). Accordingly, in the incisal view, the crown appears asymmetrical, which is further emphasized by the distal projection of a moderate lingual cervical eminence. The lingual surface is concave, bordered by moderately expressed distal and mesial marginal ridges (as also clearly shown in the EDJ, Fig 4). These ridges merge at the cervical eminence giving a semi-shovel shaped aspect to the crown (ASUDAS grade 4).

The moderate buccal bulging of the crown and the slight flaring of the mesial and distal sides from the cervix are more consistent with a lower than an upper deciduous canine. Interproximal wear facets are visible on both mesial and distal sides, the distal side being smaller (length: 1.20 mm; height: 1.73 mm) than the mesial one (length: 1.49 mm; height: 2.59 mm). The preserved root, slightly more elongated labially (mid-labial height: 2.95 mm) than lingually, (mid-lingual height: 2.31 mm) is resorbed (stage Res3/4 [45]), suggesting an age at exfoliation of approximately six years on the basis of recent human standards [38]. The crown has

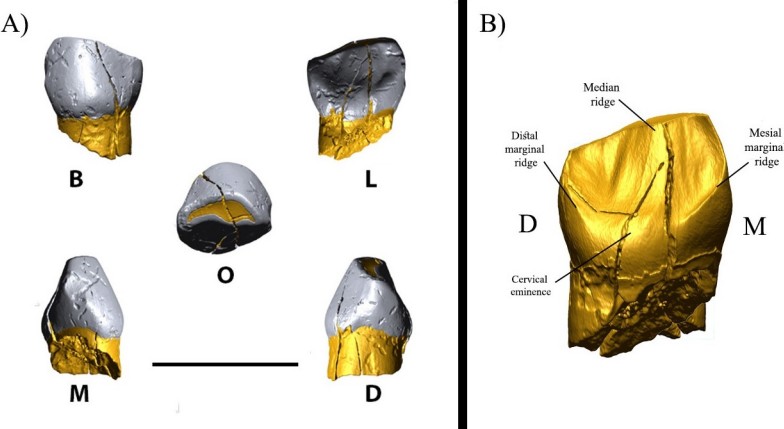

**Fig 4. Lower left deciduous canine (BY1).** A) BY1 in different views. B) The enamel-dentine junction of the tooth shows the cervical eminence and lingual ridges. Abbreviations: B: buccal, L: lingual, M: mesial, D: distal, O: occlusal.

a mesiodistal (MD) diameter of 6.87 mm (minimum estimation due to wear) and a buccolingual (BL) diameter of 6.50 mm, which is close to UP *Homo sapiens* and, particularly, Neanderthal mean values (Table 1). At the cervix, the MD and BL diameters are, respectively, 5.32 mm and 5.88 mm. Despite the reduced comparative sample and the intrinsic limits due to the wear stage of the teeth, unworn Neanderthal lower deciduous canines show the average and relative enamel thickness indices (AET and RET respectively) lower than the values observed for worn early and recent *Homo sapiens* specimens (Table 2). The Z-scores computed for the AET and RET values of the BY1 specimen fall very close to the mean values of Neanderthals at similar wear stage (i.e., Krapina 51, Tagliente 4), further supporting its Neanderthal affinity.

**Cultural materials and artefact technology and typology.** The technological and typological characteristics of the Bawa Yawan artifacts are based on detailed studies of the lithic assemblages from GHs 5 to 2. Three techno-complexes of MP Mousterian, UP "LaK" [46] and the Epipalaeolithic from Bawa Yawan have been compared with their counterparts throughout the Zagros Mountains (Figs 5–7). We use attribute analysis and chaînes opératoire approaches to compare specifically the assemblages from the West-Central Zagros sites including Warwasi and Shanidar. Key attributes include core preparation and reduction strategies, degree of core reduction, modes of flaking, platform preparations, and the retouching and re-sharpening of the tools. We recorded the technological attributes of each artefact individually and in relation to the reduction sequence in order to reconstruct the chaînes opératoires in each archaeological horizon.

**Table 1. Buccolingual (BL) crown diameter of BY1, compared to mean values of Neanderthal (NEA), early *Homo sapiens* (EHS) and recent *Homo sapiens* (RHS) lower deciduous canines.**

| Specimens | N | BL (mean±SD) | Z-score |
|---|---|---|---|
| BY 1 | | 6.5 | |
| NEA | 23[a] | 6.0±0.5 | 1 |
| UPHS | 21[a] | 6.0±0.4 | 1.25 |
| RHS | 100[b] | 5.2±0.6 | 2.16 |

[a][42];
[b][40].

**Table 2. Values of the components of enamel thickness computed for Neanderthal (NEA), early *Homo sapiens* (EHS) and recent *Homo sapiens* (RHS) lower deciduous canines at different wear stages (ws).**

| Specimens | n | Wear stage | Ve | Vcdp | Sedj | 3D AET | Z-score | 3D RET | Z-score |
|---|---|---|---|---|---|---|---|---|---|
| BY 1 | | 4 | 29.15 | 79.81 | 81.52 | 0.36 | | 8.31 | |
| NEA | 4[a] | 1 | 37.32 (10.49) | 75.31 (19.26) | 86.39 (16.85) | 0.43 (0.05) | 1.4 | 10.17 (0.70) | 2.66 |
| | 2[b] | 4 | 30.65 (4.07) | 89.02 (1,34) | 85.37 (1.21) | 0.36 (0.05) | 0 | 8.05 (1.22) | -0.21 |
| EHS | 2[c] | 3/4 | 39.37 (0.35) | 77.73 (2.72) | 83.33 (4.23) | 0.47 (0.03) | 3,67 | 11.09 (0.79) | 3,52 |
| RHS | 3[d] | 2/3 | 32.58 (4.35) | 53.94 (8.73) | 64.54 (7.93) | 0.51 (0.04) | 3,75 | 13.23 (1.56) | 3,15 |

Ve: enamel volume; Vcdp: crown dentine and pulp volume; Sedj: enamel-dentine junction surface; AET: average enamel thickness; RET: relative enamel thickness;

[a]Kebara 1, Kebara KMH4, La_Quina_LQ32, Le Ferassie 8;

[b]Krapina 51, Tagliente 4;

[c]Qafzeh 12, Qafzeh 15;

[d]Ulac 81, Ulac 140, Ulac 477 (from the Department of Human Evolution, Max Planck Institute of Evolutionary Anthropology)

The excavations revealed three archaeological horizons throughout the stratigraphy, resulting in a total of 10,411 lithic artifacts (Tables 3 and 4); those artifacts that were in primary depositional context were concentrated in two dense bands in GH5 (MP) and GH2 (MP, UP and Epipalaeolithic) (Fig 3B). MP materials were recovered from GH3 and GH4, although at lower concentration levels. At the top of the sequence in the Eastern trench, close to the rockshelter wall, there is a rich Epipalaeolithic industry, similar to that reported at the Warwasi rockshelter [47], characterised mainly by geometric microliths and microburins. Following the Epipalaeolithic, the first half of GH2 in both the Eastern and Western trenches preserves UP lithics typical of the West-Central Zagros, especially in Warwasi [46]. The lithics are characterised by laminar technology. The tools mainly dominated by different kinds of twisted and retouched bladelets and end scrapers. At the bottom of GH2, and throughout GH3 to GH5, in both the Eastern and Western trenches, lithics typical of the MP Zagros techno-complex, namely the Zagros Mousterian [48], were uncovered (Tables 3 and 4, Figs 5 and 6).

**Chronology.** Thirteen charcoal samples were collected from different parts of the Bawa Yawan stratigraphy: (a) six charcoal samples were obtained from the below and above the hominin tooth in GH4 and GH5; (b) six samples from squares I29 and G27 from the beginning and middle of the UP occupation in GH2 and GH3; (Fig 8) (c) and one sample from the Epipalaeolithic layer at the upper part of GH2 in the Eastern Trench.

Eight of the twelve samples were successfully dated, while one returned a minimum date (Tables 5 and 6). All the ${}^{14}$C results are calibrated using the new IntCal20 [43] within the OxCal 4.4 program [43, 49] and are discussed here at 68.2% probability. The youngest occupation of the site, corresponding to the Epipalaeolithic level, ranging between 13,400 and 13,300 cal. BP (Fig 9) Only two of the six charcoal samples from the UP GH2 were dated successfully (both from square G27: MAMS-41723 and MAMS-41725). The two ages obtained are consistent with each other, yielding calibrated age ranges of 34,700–34,400 and 34,600–34,400 cal. BP, respectively (Fig 3B and Table 5). Five of the six charcoal samples collected from MP occupation layers (GH4 and GH5) produced reliable radiocarbon ages; a fifth sample produced a minimum age. (Table 7 and Fig 8). The ages obtained are consistent with their respective depths within the stratigraphy. The sample from the middle part of GH4 (MAMS-44523), 30 cm above the level of the BY1 in the same square (I31), provided an age of 40,300–39,400 cal. BP. Another sample from square I31 (MAMS-44525), at 9 cm above BY1, provided an age between 42,100 and 41,800 cal. BP, and a charcoal sample at a level just 2 cm above BY1 in square H31 (Beta-465001) provides the closest age estimation of the specimen, ranging between 43,000 and

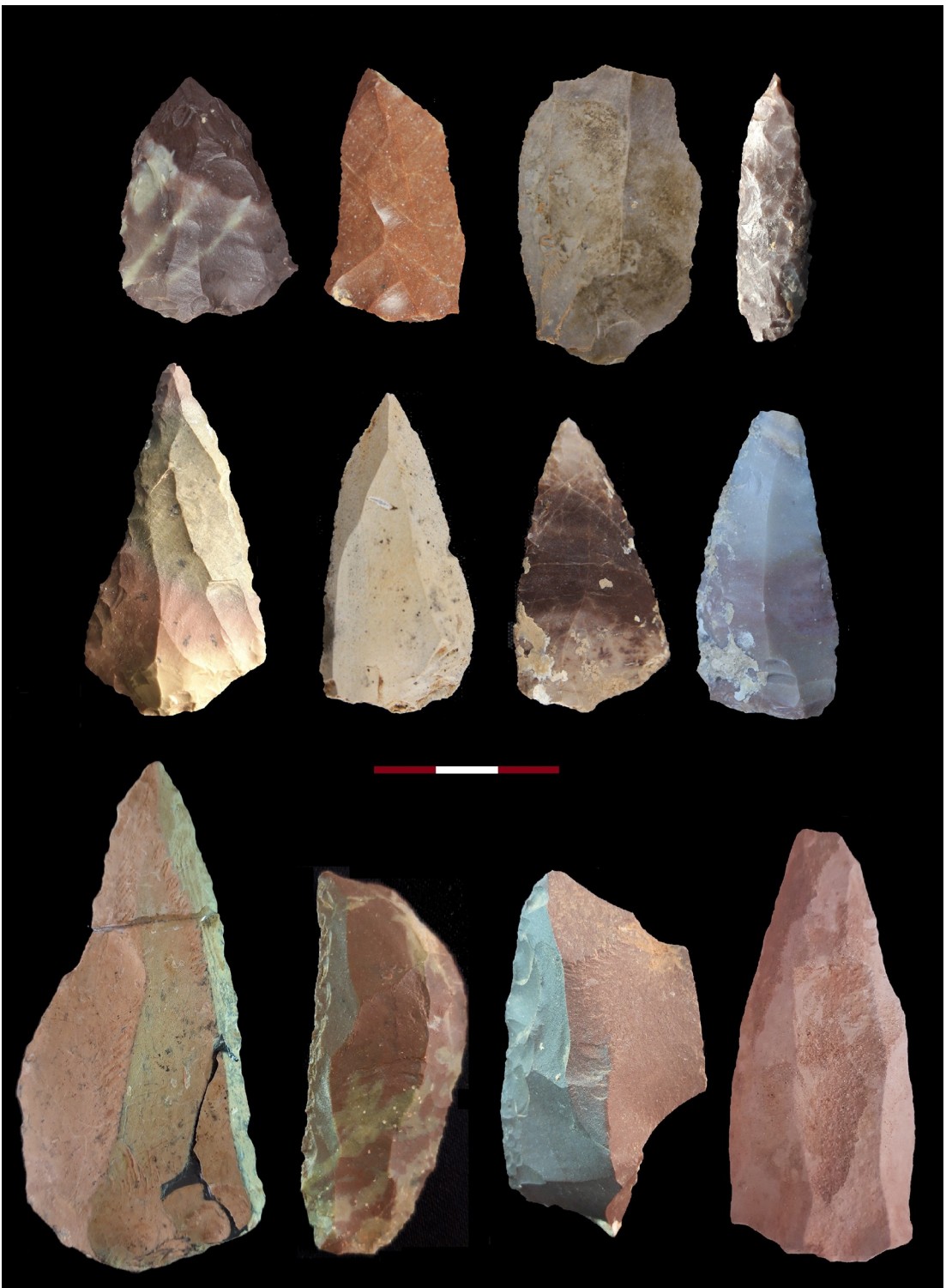

**Fig 5.** 1, 5, 7–9. Convergent scraper, 2.3.6,12. Levallois flake, 4. Limace, 10. Double scraper, 11. Single scraper.

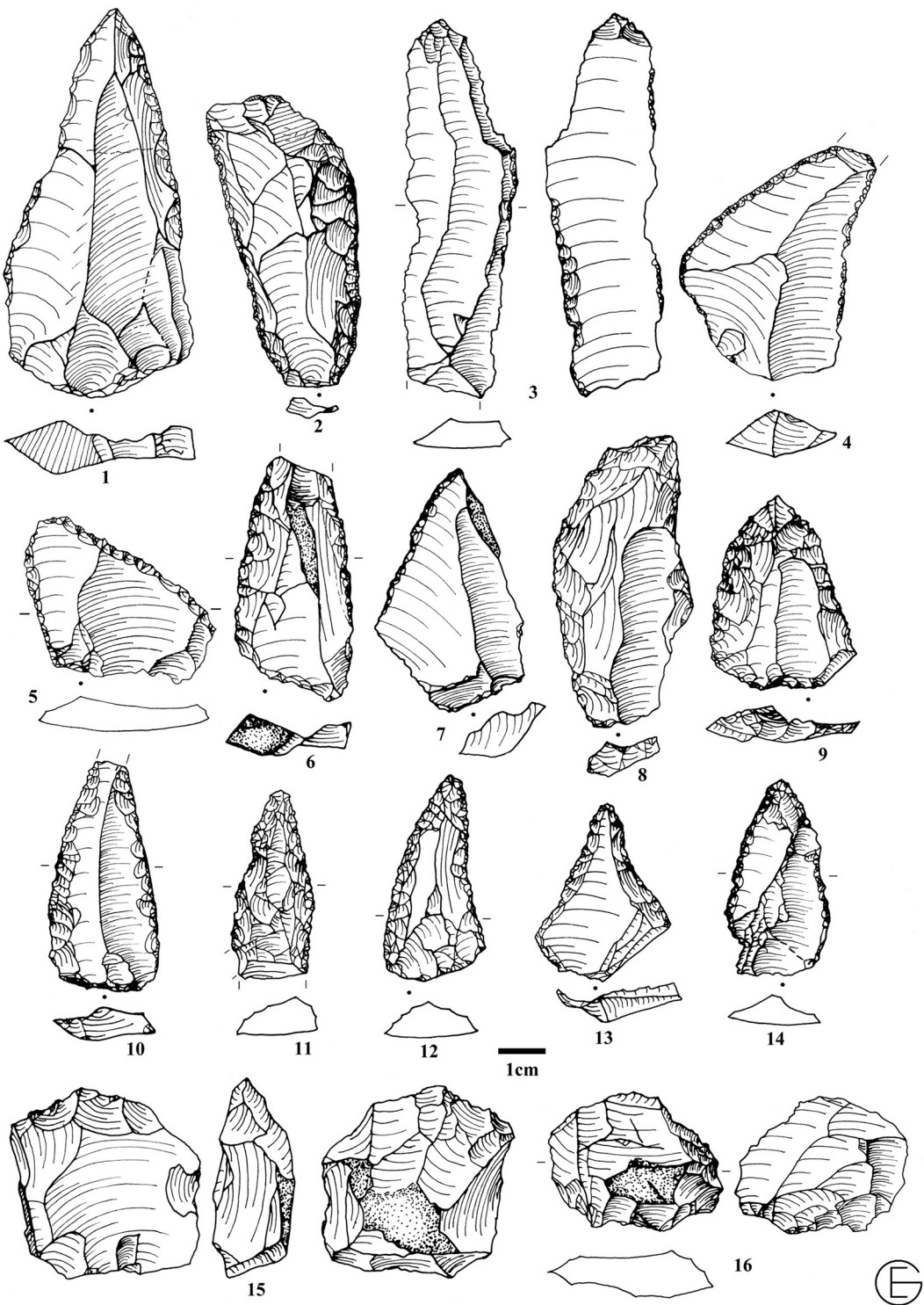

**Fig 6. A sample of the Zagros Mousterian artifices from the Bawa Yawan rockshelter around the BY 1.** 1, 9–14. Convergent scraper, 2,6. Double scraper, 3. Denticulate, 4, 5. Dejete scraper, 7. Single scraper, 8. Scraper, 15, 16. Truncated facetted.

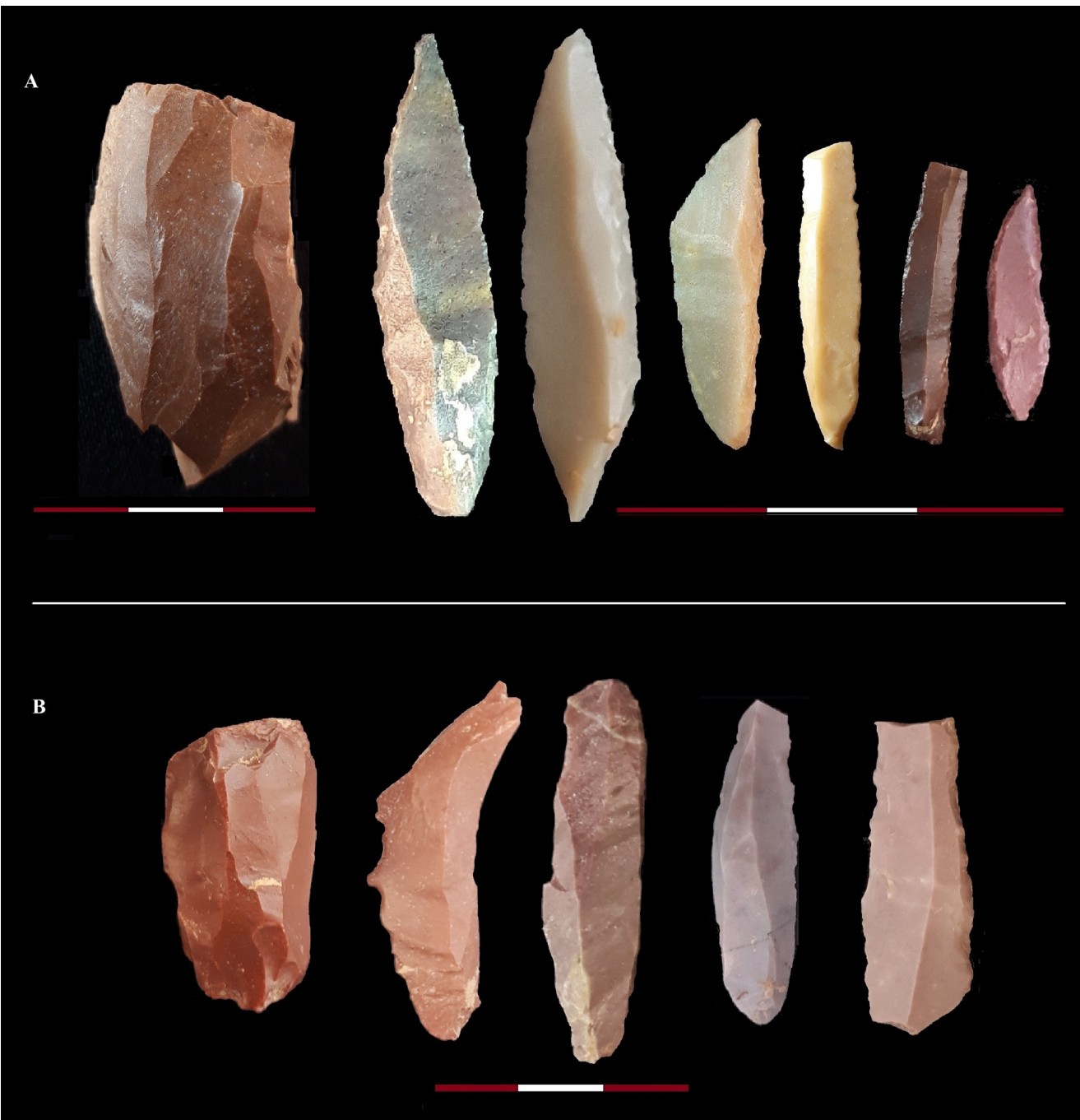

**Fig 7.** A: 1. Platform bladelet core, 2, 3. points on bladelet, 4, 5. Geometric microliths 6,7. Retouched bladelets. B:1. platform bladelet core, 2. Denticulated bladelet, 3, 4. Twisted bladelets, 4, 5. Retouched twisted bladelets.

42,500 cal. BP. The MAMS-41724 sample, 9 cm below the level of BY1 in square I29, provided an age range of 44,100–43,200 cal. BP, while a charcoal sample (OxA-36752) 11 cm below the level of BY1 in square H31 provided an age older than 44,200 BP dating range. (Table 7).

We performed a Bayesian model using the new IntCal20 in OxCal 4.4 and the t-type outlier analysis (prior probability set at 5%), to detect if there is any problem between the $^{14}$C ages and their relative stratigraphic position at the site (Fig 9 and Table 6). The model, which was run

**Table 3. All lithic artifacts from the west trench.**

| Z | Flake | blade | bladelet | core | tool | AD | SD/MD | sum |
|---|---|---|---|---|---|---|---|---|
| 20.8–20.89 | 2 | 2 | 1 | | 3 | | 2 | 10 |
| 20.7–20.79 | 19 | 8 | 12 | 1 | 6 | | 33 | 79 |
| 20.6–20.69 | 38 | 13 | 25 | | 7 | | 3 | 86 |
| 20.5–20.59 | 157 | 43 | 97 | 2 | 40 | 5 | 326 | 670 |
| 20.4–20.49 | 188 | 72 | 148 | 5 | 46 | 11 | 213 | 683 |
| 20.3–20.39 | 49 | 23 | 42 | 1 | 11 | 1 | 53 | 180 |
| 20.2–20.29 | 68 | 27 | 40 | | 12 | 4 | 29 | 180 |
| 20.1–20.19 | 114 | 7 | 32 | | 7 | 4 | 25 | 189 |
| 20.0–20.09 | 75 | 5 | 16 | | 2 | 2 | | 100 |
| 19.9–19.99 | 9 | 1 | 5 | | | | 8 | 23 |
| 19.8–19.89 | 8 | 3 | 3 | | | 1 | 14 | 29 |
| 19.7–19.79 | 4 | 1 | 7 | | 1 | | 13 | 26 |
| 19.6–19.69 | 2 | | 1 | | | | | 3 |
| 19.5–19.59 | | 1 | 1 | | | | | 2 |
| **Sum** | 733 | 206 | 430 | 9 | 135 | 28 | 719 | 2260 |

The lithic artifacts are divided into technological categories based on artificial 10 cm spit. AD: angular debris, SD: small debitage (> 10mm), MD: micro debitage (> 5mm).

using only the dates from GH5 until the GH2, confirm the integrity of the chronology with an Agreement index of 107%, well above 60% (Table 7 and Fig 8). The layer GH5 start at 44,600–43,100 ending in a transitional boundary with the GH4 at 42,000–40,500 cal. BP. In reality, one $^{14}$C age at the bottom of the sequence, result in an age older than 44,200 $^{14}$C BP, hence it could be older than the ages provided by the Bayesian model. Using the 'date' command in OxCal we estimated the duration of each phase at the site, and knowing the exact stratigraphic depth of the Neanderthal tooth in relation with all the others samples, an indirect date of BY1 could have been determined (see Method section). The total duration of GH5 ranges between 43,600 and 41,500 cal. BP, the duration of GH4 is between 41,100 and 39,000 cal. BP, with the upper part (GH2) ranging between 34,900 and 34,200 cal. BP (Table 7).

These dates indicate that the Bawa Yawan rockshelter contains a sequence from the MP to the UP in both the Western and Eastern trenches, as well as an Epipalaeolithic occupation level above those in the Eastern trench, which was not modelled. According to the Bayesian age modelling, GH4 and GH3 are broadly simultaneous with a substantially cold period in the Northern Hemisphere—the so-called Heinrich Stadial 4 (Fig 8), which might be responsible for the reduced cultural materials in this GH.

Furthermore, at this stage of research, these ranges show two critical ages for the Neanderthal occupation of the Zagros Mountains—first, the age of the Neanderthal BY1 tooth can be bracketed between 43,600 and 41,500 cal. BP, and second, the possible age of a young Zagros Mousterian in the area around 39,000 cal. BP. It is important to stress that more radiocarbon dates are on the way to help in this interpretation since the young age in GH2 is based only on a single sample.

## Discussion

On current evidence, we can identify multiple events of Neanderthal local extinctions—first, a Neanderthal group, unrelated to later local populations, becomes extinct in the Altai during Marine Isotope Stage (MIS) 5 (130,000–74,000 BP) [11, 12], second, the extinction of a

**Table 4. All lithic artifacts from the East trench.**

| Z | Flake | blade | Bladelet | Levallois blade | Levallois flake | core | tool | AD | SD/MD | sum |
|---|---|---|---|---|---|---|---|---|---|---|
| 19.4–19.49 | 4 | 1 | | | | | | | 4 | 9 |
| 19.3–19.39 | | | | | | | | | | 0 |
| 19.2–19.29 | | | | | | | | | | 0 |
| 19.1–19.19 | 11 | | 7 | | | 1 | 6 | 1 | 3 | 29 |
| 19.0–19.09 | 31 | 2 | 10 | | | | 13 | 1 | 30 | 87 |
| 18.9–18.99 | 69 | 5 | 16 | | | 3 | 29 | 3 | 29 | 154 |
| 18.8–18.89 | 69 | 13 | 23 | | | 3 | 32 | 2 | 33 | 175 |
| 18.7–18.79 | 80 | 19 | 32 | | | 4 | 15 | 4 | 28 | 182 |
| 18.6–18.69 | 135 | 16 | 57 | | | 3 | 22 | 7 | 42 | 282 |
| 18.5–18.59 | 136 | 1 | 67 | | | 8 | 20 | 7 | 61 | 300 |
| 18.4–18.49 | 230 | 37 | 71 | | | 7 | 27 | 13 | 194 | 579 |
| 18.3–18.39 | 164 | 25 | 48 | 1 | | 5 | 19 | 3 | 136 | 401 |
| 18.2–18.29 | 159 | 20 | 34 | | 1 | 8 | 19 | 3 | 149 | 393 |
| 18.1–18.19 | 205 | 16 | 30 | | 1 | 6 | 23 | 10 | 135 | 426 |
| 18.0–18.09 | 172 | 17 | 18 | | | 4 | 10 | 8 | 128 | 357 |
| 17.9–17.99 | 149 | 15 | 15 | | | 3 | 21 | 5 | 105 | 313 |
| 17.8–17.89 | 106 | 5 | 10 | | | 4 | 16 | 9 | 67 | 217 |
| 17.7–17.79 | 70 | 8 | 11 | | | | 18 | 2 | 46 | 155 |
| 17.6–17.69 | 59 | 4 | 3 | | | 2 | 6 | 3 | 29 | 106 |
| 17.5–17.59 | 28 | 2 | 1 | | | | 2 | | 11 | 44 |
| 17.4–17.49 | 14 | 4 | 1 | | 1 | | 8 | 2 | 13 | 43 |
| 17.3–17.39 | 7 | | | | | | 1 | | 4 | 12 |
| 17.2–17.29 | 10 | | | | | | 4 | 2 | 7 | 23 |
| 17.1–17.19 | 11 | | 1 | | | | 2 | | 7 | 21 |
| 17.0–17.09 | 8 | | 1 | | | | 3 | | 2 | 14 |
| 16.9–16.99 | 12 | | 0 | 1 | | | 2 | | 9 | 24 |
| 16.8–16.89 | 5 | | 2 | | | | | | 4 | 11 |
| 16.7–16.79 | 1 | | | | | | 3 | 1 | 6 | 11 |
| 16.6–16.69 | 21 | | 1 | | | | 1 | | 14 | 37 |
| 16.5–16.59 | 14 | | 1 | | | | 1 | | 10 | 26 |
| 16.4–16.49 | 19 | 2 | | | | | 2 | 3 | 28 | 54 |
| 16.3–16.39 | 48 | 3 | | | | | 2 | 4 | 80 | 137 |
| 16.2–16.29 | 41 | 3 | 1 | | | 1 | 5 | | 35 | 86 |
| 16.1–16.19 | 12 | | 5 | | | | 5 | 1 | 9 | 32 |
| 16.0–16.09 | 10 | | 1 | | | | | 1 | 4 | 16 |
| 15.9–15.99 | 6 | 1 | | | | | | | 5 | 12 |
| 15.8–15.89 | 39 | 1 | 2 | | 2 | 2 | 10 | 4 | 48 | 108 |
| 15.7–15.79 | 86 | 1 | 4 | 1 | 2 | | 8 | 3 | 73 | 178 |
| 15.6–15.69 | 188 | 7 | 4 | | 2 | 2 | 27 | 2 | 261 | 493 |
| 15.5–15.59 | 152 | 10 | 15 | 2 | 4 | 1 | 21 | 10 | 255 | 470 |
| 15.4–15.49 | 175 | 7 | 6 | 1 | 4 | 1 | 26 | 8 | 153 | 381 |
| 15.3–15.39 | 249 | 6 | 12 | | 5 | 1 | 35 | 9 | 173 | 490 |
| 15.2–15.29 | 241 | 9 | 13 | 1 | 10 | | 24 | 11 | 111 | 420 |
| 15.1–15.19 | 196 | 5 | 5 | | 5 | | 29 | 6 | 78 | 324 |
| 15.0–15.09 | 50 | 2 | 1 | 1 | 2 | | 10 | 1 | 28 | 95 |
| 14.9–14.99 | 55 | 2 | 1 | | 1 | 1 | 9 | 5 | 17 | 91 |
| 14.8–14.89 | 40 | 2 | | | | | 5 | 3 | 15 | 65 |

*(Continued)*

**Table 4.** (*Continued*)

| Z | Flake | blade | Bladelet | Levallois blade | Levallois flake | core | tool | AD | SD/MD | sum |
|---|---|---|---|---|---|---|---|---|---|---|
| 14.7–14.79 | 30 | 2 | | | | 1 | 12 | | 13 | 58 |
| 14.6–14.69 | 55 | 3 | 1 | 1 | | 2 | 10 | | 14 | 86 |
| 14.5–14.59 | 52 | 2 | 1 | 2 | 1 | 1 | 16 | | 12 | 87 |
| 14.4–14.49 | 11 | | 1 | | | | 2 | 1 | 11 | 26 |
| 14.3–14.39 | 1 | | | | | | 1 | | 2 | 4 |
| 14.15–14.3 | 2 | | | | | | | | 1 | 3 |
| **Sum** | **3738** | **278** | **533** | **11** | **41** | **74** | **582** | **158** | **2732** | **8151** |

The lithic artifacts are divided into technological categories based on artificial 10 cm spit. AD: angular debris, SD: small debitage (> 10mm), MD: micro debitage (> 5mm).

different population of Neanderthals in the Altai between 59,000 and 49,000 BP [11, 12] that may overlap with the local disappearance of a Neanderthal population in the Crimea [50]; third, the disappearance of Neanderthals from the Levant by 48,000–49,000 BP [20], followed by a final event of the approximately synchronous disappearance of Neanderthal populations across a vast territory—from Western Europe to the Caucasus—between 40,000 and 37,000 BP

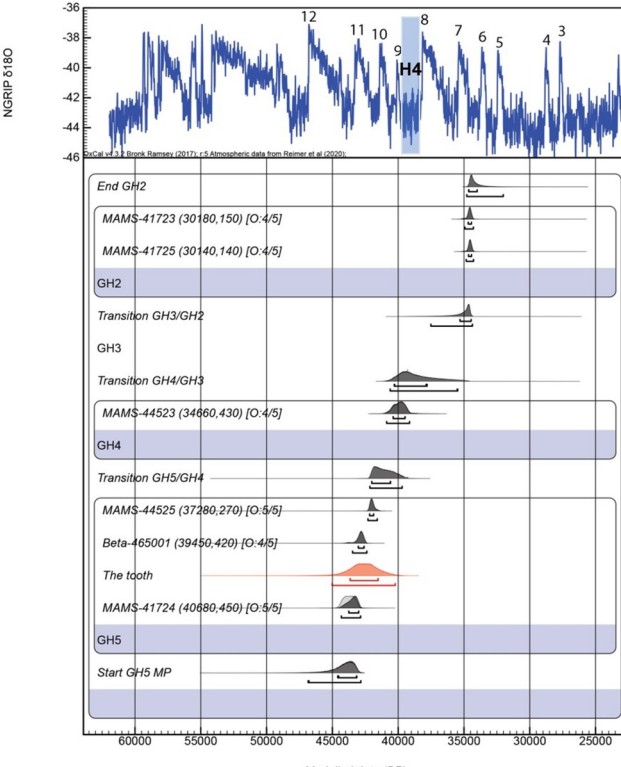

**Fig 8. Bayesian model of Bawa Yawan rockshelter.** Radiocarbon dates are calibrated using IntCal20 [43]; the model and boundaries were calculated using OxCal 4.3, including a General t-type Outlier Model [49]. Outliers prior and posterior probability are shown in square brackets. The chronology is compared to the North Greenland Ice Core Project (NGRIP) Greenland Ice Core Chronology 2005 (GICC05) $\delta^{18}$O palaeo-environmental record [50] with warm Dansgaard-Oeschger events (from 12 to 3) and Heinrich stadial 4(H4).

**Table 5. Table shows the $^{14}$C dates available for the Bawa Yawan rockshelter sequence and of BY1 tooth, with its respective position in the layer (Column Z).**

| N | Period GH | QU | Z | GH | MPI Code | Lab. Code | Calibrated $^{14}$C ages (cal. BP) | 1s Err |
|---|---|---|---|---|---|---|---|---|
| 1 | Epi | E13 | 20.47 | 2 | R-EVA 3304 | MAMS-41722 | 11516 | 37 |
| 2 | UP | I29 | 18.37 | 2 | R-EVA 3299 | MAMS-41721 | -1535 | 19 |
| 3 | UP | I29 | 18.12 | 2 | R-EVA 3308 | MAMS-41727 | 76 | 20 |
| 4 | UP | I29 | 18.09 | 2 | R-EVA 3452 | MAMS-44524 | -53 | 19 |
| 5 | UP | G27 | 18.07 | 2 | R-EVA 3292 | MAMS-41723 | 30180 | 150 |
| 6 | UP | G27 | 18.04 | 2 | R-EVA 3295 | MAMS-41725 | 30140 | 140 |
| 7 | UP | I29 | 17.77 | 3 | R-EVA 3296 | MAMS-41726 | -265 | 20 |
| **8 | MP | I31 | 16.16 | 4 | R-EVA 3451 | MAMS-44523 | 34660 | 430 |
| 9 | MP | I31 | 15.93 | 5 | R-EVA 3453 | MAMS-44525 | 37280 | 270 |
| 10 | MP | H31 | 15.88 | 5 | Beta-465001 | Beta-465001 | 39450 | 420 |
|  | BY1 | I31 | 15.86 | 5 | - | - | - | - |
| 11 | MP | I29 | 15.77 | 5 | R-EVA 3300 | MAMS-41724 | 40680 | 450 |
| 12 | MP | H31 | 15.75 | 5 |  | OxA-36752 | >44200 | - |

[2, 4, 22]. Contrasting with the synchronicity of their final disappearance, these late Neanderthal groups appear to be spatially structured and to have different demographic trajectories.

The discoveries from the Bawa Yawan rockshelter bring clarity to the position of the Zagros Mountains in this emerging picture. The presence of Neanderthals in the Zagros has been known for decades [26, 27, 29, 32]. However, few sites (Shanidar, Warwasi, Ghar-e-Khar, Kaldar, Ghamari, Gilvaran) preserve both MP and UP industries [25–27, 51, 52], while chronometric ages derive only from the old and new excavations at Shanidar, which point to the use of the cave by Neanderthals between 70,000 and 45,000 BP [26–28]. The age between 43,000 and 41,000 cal. BP for BY1 tooth associated with the Zagros Mousterain artifcts in the West-

**Table 6. Bayesian Modelled calibrated ages and boundaries of Model 1 provided by the IntCal20 using OxCal 4.3 program [43, 49].** In red is the indirect date of the BY1 tooth provided by the 'date' command in OxCal.

| Bawa Yawan rockshelter | Un-Modelled (BP) | | | | Modelled (BP) | | | |
|---|---|---|---|---|---|---|---|---|
| Indices Amodel 108.2 Aoverall 107 | cal. BP 68.2% | | cal. BP 95.4% | | cal. BP 68.2% | | cal. BP 95.4% | |
|  | from | To | from | To | From | To | from | to |
| **End GH2** |  |  |  |  | 34650 | 34000 | 34790 | 32010 |
| MAMS-41723 (30180;150) | 34700 | 34410 | 35050 | 34290 | 34680 | 34440 | 34940 | 34280 |
| MAMS-41725 (30140;140) | 34660 | 34400 | 34900 | 34240 | 34660 | 34430 | 34840 | 34270 |
| GH2 |  |  |  |  |  |  |  |  |
| **Transition GH3/GH2** |  |  |  |  | 35310 | 34460 | 37520 | 34350 |
| GH3 |  |  |  |  |  |  |  |  |
| **Transition GH4/GH3** |  |  |  |  | 40300 | 37800 | 40610 | 35470 |
| MAMS-44523 (34660;430) | 40310 | 39400 | 40860 | 39080 | 40390 | 39470 | 40880 | 39130 |
| GH4 |  |  |  |  |  |  |  |  |
| **Transition GH5/GH4** |  |  |  |  | 42010 | 40570 | 42150 | 39700 |
| MAMS-44525 (37280;270) | 42160 | 41850 | 42270 | 41570 | 42180 | 41870 | 42300 | 41580 |
| Beta-465001 (39450;420) | 43050 | 42590 | 43850 | 42400 | 43030 | 42590 | 43470 | 42380 |
| BY1 |  |  |  |  | 43660 | 41520 | 45030 | 40220 |
| MAMS-41724 (40680;450) | 44150 | 43280 | 44460 | 43000 | 43750 | 43000 | 44330 | 42850 |
| GH5 |  |  |  |  |  |  |  |  |
| **Start GH5 MP** |  |  |  |  | 44600 | 43140 | 46850 | 42830 |

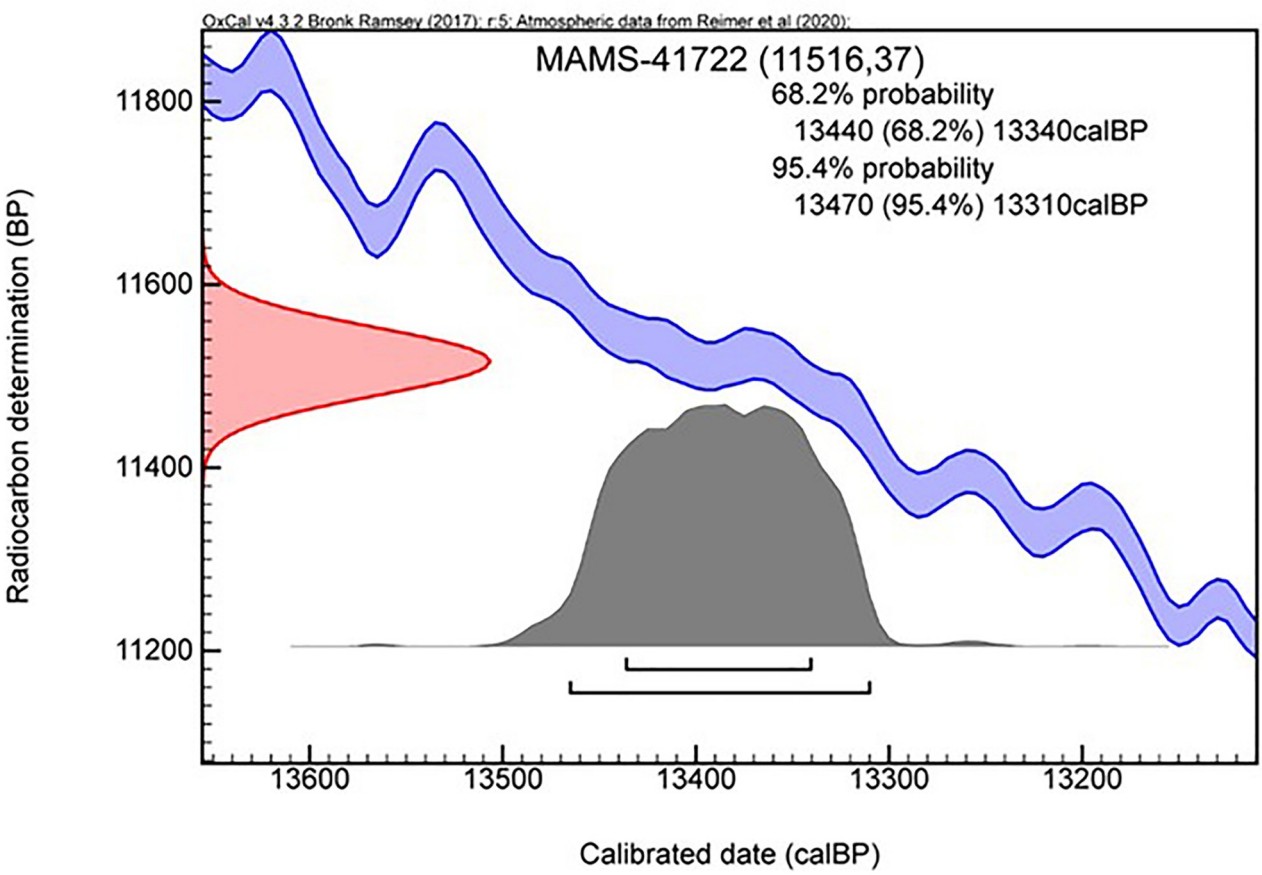

**Fig 9. Calibrated radiocarbon date of Epipaleolithic layer in eastern trench of the Bawa Yawan rockshelter used IntCal20 in OxCal 4.3 program.**

Central Zagros revealed the presence of the Neanderthals at least until around 41,000 cal. BP. The Zagros Mousterian techno-complex continues above BY1 in GHs 4 and 3. However, in the lack of any hominin physical remains, yet it is not possible to evaluate the hominin type in these GHs and this remains an open hypothesis awaits further findings. The new data of Bawa

**Table 7. Bayesian modelled boundaries and duration of the phases provided by the IntCal20[13] using OxCal 4.3 program [49].**

|  | Modelled (BP) | | | |
|---|---|---|---|---|
|  | cal. BP 68.2% | | cal. BP 95.4% | |
|  | from | To | From | To |
| End GD2 | 34650 | 34000 | 34790 | 32010 |
| **Total Duration of GH2** | **34960** | **34220** | **36620** | **33000** |
| Transition GD3/GD2 | 35310 | 34460 | 37520 | 34350 |
| Transition GD4/GD3 | 40300 | 37800 | 40610 | 35470 |
| **Total Duration of GH4** | **41110** | **39090** | **41840** | **37320** |
| Transition GD5/GD4 | 42010 | 40570 | 42150 | 39700 |
| Transition GD5/GD4 | 42010 | 40570 | 42150 | 39700 |
| **Total Duration of GH5** | **43670** | **41540** | **45020** | **40210** |
| Start GD5 MP | 44600 | 43140 | 46850 | 42830 |

Yawan together with the information from southern Caucasus [21, 23] suggest that caves and rockshelters throughout the entire mountains arch that extends from Georgia to Iran were used by relict population of Neanderthals could survived a short time after they had become extinct across most of Europe.

Furthermore, similar to the records of the Levant and Europe, the Zagros early UP assemblages are techno-typologically diverse, showing variable patterns of similarities and differences to early UP industries elsewhere, and suggesting a diversity of modern human groups with varying foraging strategies and ranges along the vast mountain chain [53]. The Bawa Yawan discoveries hypothesizes that the final disappearance of Neanderthals took place in the context of increasing competition with *Homo sapiens*, the multiple events of localised Neanderthal extinction, together with increasing genomic evidence of comparatively smaller population sizes in marginal areas [11]. The combination of these differences in demographic resilience between the two groups during periods of repeated climatic insults [54, 55], coupled with the more sophisticated technology of expanding modern humans (e.g.,[56, 57] may have been the tipping point that created what appears as a vast spatial extinction event across thousands of kilometres, but was the sum of numerous local independent population processes, among which the disappearance of Neanderthals in the Zagros Mountains was one of the last. Finally, it will always be the case that artefactual data will be far more abundant than fossil evidence. Yet, as recent work in Central Asia and Siberia has shown [12], there are localised regional signatures among MP industries that may reflect cultural traditions associated with particular populations.

The sediment characteristics of this GH suggest a rather warm climate (comparing to the other GHs) leading to a more diverse fauna to appear in the environment. Unfortunately, due to the lack of high-resolution, well-dated paleoclimate records from the Zagros Mountains for the past 45,000 years, correlating our GHs to past regional climate changes is difficult. However, pollen evidence from Lake Zeribar characterised by high percentages of *Artemisa* and Chenoppodiaceae indicates the dominance of a cool, dry steppe for the period between 40,000 and 14,000 cal. BP. Nevertheless, a slight increase in tree pollen at around 40,000 BP suggests a fairly warmer and wetter (less cold) climate at this time [58], which may correspond to the upper part of GH5.

Sediment compositions of GH4 and GH3 suggesting a prominent cold period characterized by higher snowfall simultaneous with Heinrich Stadial 4 (Figs 3B and 3C and 9). The highly calcareous nature of the sediments might point to an increase in the rate of chemical weathering due to the contact of snow meltwater with the surrounding limestone. One might speculate that the discharge of meltwater runoff rich in calcium carbonate into the rockshelter deposits has caused the highly calcareous sediments here. Moreover, the observed limestone fragments might originate from physical glacial erosion and permafrost degradation. Several lines of evidence such as a deep depression of the snowline to an elevation of about 1800 m in the Zagros Mountains during glacial periods [59], the enhanced rate of calcium carbonate deposition in the nearby Hashilan Wetland during cold climate intervals [60], and extremely sensitive climate of the West Asia to Heinrich cold events [61] provide supporting evidence for our interpretation that the observed highly calcareous sediments in this GH are associated with cold climatic conditions during Heinrich Stadial 4. Pollen evidence from Lake Zeribar reflects an almost treeless environment for the Zagros Mountains between ca. 33,000 and 15,000, which has been attributed to a colder and drier climate [58]. Given a considerable time gap at the transition from GH2 to GH1, it can be postulated that an intervening horizon has most likely been disappeared due to human interferences in the rockshelter environment.

The MP-UP transition in Europe which started between 47 kya BP overlapped with the spread of *Homo sapiens* and Neanderthals around 40 kya. Recent Palaeogenetic studies show

that the gene flow between Neanderthals and *Homo sapiens* have occurred in roughly between 60–50 probably in southwestern Asia [62, 63]. This new information is consistent with the data from Bawa Yawan. The Bawa Yawan Neanderthal suggests an extirpation date of around 45,000–40,000 cal. BP, but the lithic evidence shows extend for a while. If the conditions under which Neanderthals became extinct, locally and overall, are to be determined, it is necessary to develop a more integrated approach to the problem.

## Conclusion and implications

The discovery of an *in situ* Neanderthal tooth at the Bawa Yawan rockshelter in the West-Central Zagros Mountains, its direct association with typical Zagros Mousterian artifacts, and the chronological attempt of the local MP to UP transition, has implications for our understanding of the spatial patterning of Neanderthal extinction and its relationship to expanding *Homo sapiens* populations and further work need to be done in order to shed more light on this intriguing region. In addition to expose a sequence of three archaeological techno-complexes of MP, UP and Epipalaeolithic formed in four GHs, excavations in Bawa Yawan rockshelter offer evidence that Neanderthals had a demographically dynamic history, shaped by multiple dispersal events and localised population extinctions. Since there is reliable evidence for diverse *Homo sapiens* in the Middle East during past 45,000, the last Zagros Neanderthals potentially overlapped with groups in space and time.

## Acknowledgments

We are grateful to T. Higham, K. Douka, who kindly provided radiocarbon dating (sample number OxA-36752) for this project, and for useful discussion on the paper. We thank to a number of people who assisted us on this project late M. Azarnoosh, Y. Moradi, H. Chubak, A. Tahmasebi, F. Biglari, R. Shirazi (Iranian Center for Archaeological Research), B. Omrani (Research Institute of Cultural Heritage and Tourism), A. Gillies, K. Janin, N. Hyedari, A. Tahmasebi, S. Kermajani, G.Weniger, R. Azizi, N. Naderian, E. Fotuhi, more specifically the excavation crew, the Heydari family and the local residents of Yawan village.

## Author Contributions

**Conceptualization:** Saman Heydari-Guran.

**Data curation:** Saman Heydari-Guran, Stefano Benazzi, Sahra Talamo, Nemat Hariri, Robert A. Foley.

**Formal analysis:** Saman Heydari-Guran.

**Funding acquisition:** Saman Heydari-Guran.

**Investigation:** Saman Heydari-Guran, Stefano Benazzi, Sahra Talamo, Elham Ghasidian, Nemat Hariri, Faramarz Azizi, Rahmat Naderi, Reza Safaierad, Robert A. Foley.

**Methodology:** Saman Heydari-Guran, Stefano Benazzi, Sahra Talamo, Elham Ghasidian, Gregorio Oxilia, Jean-Jacques Hublin.

**Project administration:** Saman Heydari-Guran.

**Resources:** Saman Heydari-Guran.

**Software:** Saman Heydari-Guran, Stefano Benazzi, Sahra Talamo, Gregorio Oxilia.

**Supervision:** Saman Heydari-Guran.

**Validation:** Saman Heydari-Guran.

**Visualization:** Saman Heydari-Guran, Sahra Talamo, Gregorio Oxilia, Samran Asiabani, Faramarz Azizi.

**Writing – original draft:** Saman Heydari-Guran, Stefano Benazzi, Sahra Talamo, Elham Ghasidian, Reza Safaierad, Robert A. Foley, Marta M. Lahr.

**Writing – review & editing:** Saman Heydari-Guran, Stefano Benazzi.

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
