## [Decision Letter · Decision Letter 0]

24 Mar 2021

PONE-D-21-00463

Main Manuscript for The discovery of an in situ Neanderthal remain in the Bawa Yawan Rockshelter, West-Central Zagros Mountains, Kermanshah

PLOS ONE

Dear Dr. Heydari-Guran,

Thank you for submitting your manuscript to PLOS ONE. After careful consideration, we feel that it has merit but does not fully meet PLOS ONE’s publication criteria as it currently stands. Therefore, we invite you to submit a revised version of the manuscript that addresses the points raised during the review process.

We look forward to receiving your revised manuscript.

Kind regards,

David Caramelli, Ph.D

Academic Editor

PLOS ONE

Journal Requirements:

2. Please remove "Main Manuscript" from the title of the submission.

"The ‘Human Evolution in the Zagros Mountains (HEZM)’ project is supported by a number of

organisations including the Iranian Cultural Heritage and Tourism Organization (ICHTO). We

thank ICHTO of Kermanshah Province. The project is funded by the European Commission

H2020 (Marie Skłodowska-Curie, MUP-Trans Zagros project, grant number 702130), the British

Institute for Persian Studies (BIPS) and Deutsch Forschungsgemeinschaft (DFG), project

number 423897519 (https://gepris.dfg.de/gepris/projekt/423897519). S. Talamo is supported by

the European Research Council under the European Union’s Horizon 2020 Research and

Innovation Programme (grant agreement No. 803147 RESOLUTION,

https://site.unibo.it/resolu47Ation-erc/en). S. Benazzi and G. Oxilia are founded by the European

Research Council under the European Union’s Horizon 2020 Research and Innovation

Programme (grant agreement No. 724046 SUCCESS; http://www.erc-success.eu/). E.

Ghasidian funded by Deutsch Forschungsgemeinschaft (DFG), project number 414357211. "

"Specify the role played"

6. We note that Figures 1 and 3 in your submission contain map images which may be copyrighted. All PLOS content is published under the Creative Commons Attribution License (CC BY 4.0), which means that the manuscript, images, and Supporting Information files will be freely available online, and any third party is permitted to access, download, copy, distribute, and use these materials in any way, even commercially, with proper attribution. For these reasons, we cannot publish previously copyrighted maps or satellite images created using proprietary data, such as Google software (Google Maps, Street View, and Earth). For more information, see our copyright guidelines: http://journals.plos.org/plosone/s/licenses-and-copyright.

6.1.    You may seek permission from the original copyright holder of Figures 1 and 3 to publish the content specifically under the CC BY 4.0 license. 

6.2.    If you are unable to obtain permission from the original copyright holder to publish these figures under the CC BY 4.0 license or if the copyright holder’s requirements are incompatible with the CC BY 4.0 license, please either i) remove the figure or ii) supply a replacement figure that complies with the CC BY 4.0 license. Please check copyright information on all replacement figures and update the figure caption with source information. If applicable, please specify in the figure caption text when a figure is similar but not identical to the original image and is therefore for illustrative purposes only.

7. Please upload a copy of Figure 5, to which you refer in your text on page 9. If the figure is no longer to be included as part of the submission please remove all reference to it within the text.

Reviewers' comments:

Reviewer's Responses to Questions

**Comments to the Author**

1. Is the manuscript technically sound, and do the data support the conclusions?

Reviewer #1: Yes

Reviewer #2: Yes

2. Has the statistical analysis been performed appropriately and rigorously? 

Reviewer #1: N/A

Reviewer #2: Yes

3. Have the authors made all data underlying the findings in their manuscript fully available?

Reviewer #1: Yes

Reviewer #2: Yes

4. Is the manuscript presented in an intelligible fashion and written in standard English?

Reviewer #1: Yes

Reviewer #2: Yes

5. Review Comments to the Author

Reviewer #1: The ms provides interesting new information on an area of Eurasia poorly known in term of Paleolithic occupation. The ms is well structured is some sections, but not in others, and it requires some reworking.

Reviewer #2: The article entitled "The discovery of an in situ Neanderthal remain in the Bawa Yawan Rockshelter, West-Central Zagros Mountains, Kermanshah" by Heydari-Guran and colleagues, reports the discovery of a deciduous Neanderthal tooth from the Zagros Mountains region in Iran. The specimen (BY 1) has been dated to between ~43,600 and ~41,500 years ago, thus providing the most recent date for the presence of Neanderthals in the region. From this point of view, despite its fragmentary nature, BY 1 offers important clues for understanding the dynamics of the transition between the Middle to Upper Palaeolithic transition and the extinction of the Neanderthals in the Middle East and also, more generally, throughout the Central-Western regions of Eurasia. The anatomical description of the tooth provided by the authors is rigorous and its attribution to Homo neanderthalensis appears fully justified on the basis of its morphometric features and the stratigraphic position within the MP level labeled GH4. In the article, the authors provide a convincing reconstruction of the demographic and ecological events of expansion and extinction of human populations in the Middle East during the transition between MP and UP. For these reasons, the article is of definite scientific interest and I am pleased to recommend its publication pending some minor revisions.

My main perplexity concerns the chronological horizon (~40,000-37,000 cal BP) in which the authors place the disappearance of Neanderthals in Europe. The definition of this horizon, and in particular of the most recent date of 37,000, is particularly important in an article such as this one that focuses on the timing of Neanderthal extinction in comparison between different geographical areas and should, in my opinion, be better justified (please see my specific comment below for more details).

Revisions by lines:

Line 17: word mispelled

In the abstract and later in the text in lines 20-21, 39, 363 the authors report the chronological horizon ~ 40,000-37,000 cal BP as the most recent date for the disappearance of Neanderthals in Europe. This statemet is justified on the basis of articles reported in bibliography as 1-4 and 22 with particular reference to articles by Higham and colleagues (2014, ref.2), Talamo and colleagues (2020, ref. 3), and Zilhão and colleagues (2017, ref.4). However, since the recent dating of the Neanderthal fossil remains from Spy to more than 40 thousand years ago (Devièse and colleagues, 2021 doi.org/10.1073/pnas.2022466118) there is to my knowledge no Neanderthal human fossil remains that have returned a firm date less than 40 thousand years BP. The most recent dates available are based solely on archaeological levels of the late Mousterian or of the so-called transitional industries. The dates reported by Zilhão and colleagues (2017) for the Spanish site were calibrated with IntCal13 (Reimeretal., 2013) inCalib7.0.4 (Stuiver and Reimer, 1993) and are somewhat controversial. Regarding the dates reported in Talamo and colleagues (2020), it seems to me, but Talamo may contradict me, that all the dates most recent than 40,000-year ago (and in particular those at 37,000 years BP) refer to levels with transitional industries (Châtelperronian) and not to levels with final Mousterian.

Regardless of the views of the authors of this reviewed article, some of whom are among the leading proponents of the association between Chatelperronian industries and Neanderthals, the issue remains controversial and there is no unanimous agreement among scholars about such association. Thus, the 37,000 year old date for the extinction of Neanderthals in Europe only stands on the assumption that the Chatelperronian tools (of that period) were made by Neanderthals, as there is no fossil or archaeological evidence for the final Mousterian more recent than 40,000 years ago. So, about the date of 37,000 years BP as the LAD for the Neanderthals I think that the autors should specify more clearly in text to which industries they are referring to and of the evidences of their possible association with Neanderthals. This justification would not be necessary if the focus the authors themselves have given to the article was not specifically on the dynamics of Neanderhal extinction and the comparison between different geographical areas including Europe.

lines 191 and 212: only genus and species names should be italicised, not the names of taxonomic families.

line 192: Please add "excavation" after "seasons"

Lines 340 and 341: on the basis of the dates and descriptions provide for levels GH3-Gh5 the most recent date of 39 thousand years for the Mousterian in the region seems to be not "probable" (as reported in line 340) but only "possible".

The article is an important contribution to the knowledge of the Neanderthals in the East and about the population and demographic dynamics and the tempo and mode of their extinction. I thank the editor for giving me the opportunity to review this fine article. I have no corrections to point to for the SI. I am not an anonymous referee.

6. PLOS authors have the option to publish the peer review history of their article (what does this mean?). If published, this will include your full peer review and any attached files.

Reviewer #1: No

Reviewer #2: **Yes: **Fabio Di Vincenzo

---

## [Author Response · Author response to Decision Letter 0]

14 May 2021

To whom it may concern,

We have made the quality of figure 3 much much higher. We have also brought almost all the supplementary information to the main text. We have accepted the comments of reviewers and applied them in the manuscript. 

sincerely yours 

Saman

---

## [Editor Report · Decision Letter 1]

11 Jun 2021

Main Manuscript for The discovery of an in situ Neanderthal remain in the Bawa Yawan Rockshelter, West-Central Zagros Mountains, Kermanshah

PONE-D-21-00463R1

Dear Dr. Heydari-Guran,

We’re pleased to inform you that your manuscript has been judged scientifically suitable for publication and will be formally accepted for publication once it meets all outstanding technical requirements.

Kind regards,

David Caramelli, Ph.D

Academic Editor

PLOS ONE
---

## [Editor Report · Acceptance letter]

16 Aug 2021

PONE-D-21-00463R1 

The discovery of an *in situ* Neanderthal remain in the Bawa Yawan Rockshelter, West-Central Zagros Mountains, Kermanshah 

Dear Dr. Heydari-Guran:

I'm pleased to inform you that your manuscript has been deemed suitable for publication in PLOS ONE. Congratulations! Your manuscript is now with our production department. 

Kind regards, 

on behalf of

Professor David Caramelli 

Academic Editor

PLOS ONE